# Quality Characteristics of Sweet Potato Jelly Prepared Using the Enzymatic Saccharification Method

**DOI:** 10.3390/foods12081585

**Published:** 2023-04-08

**Authors:** Hye-Won Lee, Yeon-Jae Jo, Yun-Jo Jung, Mi-Nam Chung, Jun-Soo Lee, Heon-Sang Jeong

**Affiliations:** 1Department of Food Science and Biotechnology, Chungbuk National University, Cheongju 28644, Republic of Korea; wjq63639@naver.com (H.-W.L.); yjcho6522@naver.com (Y.-J.J.); yunjo96@naver.com (Y.-J.J.); junsoo@chungbuk.ac.kr (J.-S.L.); 2Bioenergy Crop Research Institute, Rural Development Administration, Muan 58545, Republic of Korea; minam@korea.kr

**Keywords:** sweet potato, jelly, enzyme hydrolysis, quality characteristics, sensory evaluation

## Abstract

In this study, jelly was prepared using saccharified sweet potatoes without sugar, and its quality characteristics were compared according to the sweet potato cultivar. Three sweet potato varieties, namely Juwhangmi (orange color), Sinjami (purple color), and Daeyumi (yellow flesh color), were used. The total free sugar and glucose contents of the hydrolysate were found to increase during the enzyme treatment. However, no differences in the moisture, total soluble solids, or textural properties were found among the sweet potato cultivars. Sinjami had high total polyphenol and flavonoid contents of 446.14 mg GAE/100 g and 243.59 mg CE/100 g, respectively, and it had the highest antioxidant activity among the cultivars. Based on the sensory evaluation, an overall preference appeared in the order of Daeyumi, Sinjami, and Juwhangmi cultivars. This result shows that jelly can be manufactured by saccharifying sweet potatoes, and it was confirmed that the characteristics of raw sweet potatoes had a great influence on the quality characteristics of the jelly. Further, the characteristics of raw sweet potatoes had a remarkable influence on the quality characteristics of the jelly.

## 1. Introduction

Sweet potato (*Ipomoea batatas* L.) is an edible dicotyledonous crop that belongs to the Convolvulaceae family. Sweet potato is an excellent source of carbohydrates and a major food resource in Korea, in addition to rice and soybeans. As the world’s seventh largest crop, sweet potato is widely used as a functional food owing to its high content of carbohydrates, phenolic compounds, carotenoids, vitamin C, anthocyanins, various minerals, and dietary fiber [1]. In addition, sweet potato has attracted remarkable attention as a crop that simultaneously solves the energy, food, and environmental problems of the 21st century owing to its strong environmental adaptability and can consume its roots, stems, and leaves [2]. Sweet potatoes have a high utility value as they are used as food, snacks, processed foods, vegetables, and feed, and they can also be applied for industrial use [3]. Sweet potatoes are composed of 70% moisture, 26.4% carbohydrates, 1.8% protein, and 0.6% fat, and they contain active ingredients such as polyphenols, yalapine, and gangliosides [4]. As new sweet potato cultivars with improved functionality have been developed and produced, these cultivars have attracted attention as functional foods with nutritional properties and functionality [5].

Jelly, a gel-state food, is a saccharide favorite containing approximately 20% water. Depending on the type of gelling agent that can bind moisture, a diverse texture can be obtained, and various products can be expected depending on the manufacturing process [6]. The general manufacturing process for jelly involves mixing a gelling agent and saccharides, followed by a series of concentration, molding, and drying [7]. Jelly is a semi-solid food that is widely used because of its soft texture and easy swallowing. As interest in the texture of food is also increasing, jelly consumption as a dessert is increasing [8,9]. Owing to a recent increase in women’s interest in health and beauty, research has been conducted on various functional stick-type jellies that are easy to carry and release via the addition of sub-materials rich in antioxidants [10].

Sweet potato has a higher yield per unit area than grains, such as rice, wheat, and corn; therefore, it is highly usable as a raw material for processing. It is also the seventh largest carbohydrate-supplied food in the world. Sweet potatoes are important crops in terms of nutrition and cultivation, and continuous research on various processing methods that utilize sweet potatoes is necessary. In this study, by increasing the sugar content of sweet potatoes through enzyme treatment, jelly was prepared using only saccharified sweet potatoes without the addition of sugar, and the quality characteristics of jelly prepared from different sweet potato varieties were compared.

## 2. Materials and Methods

### 2.1. Materials

Sweet potatoes (*Ipomoea batatas* L.) with differences in flesh color were harvested in October 2021 by the Rural Development Administration’s Bio Energy Center. These cultivars include Juwhangmi (orange color), Sinjami (purple color), and Daeyumi (yellow flesh color). The Daeyumi cultivar was used as the control because of its excellent taste and flesh color, which represents the common sweet potato. The harvested sweet potatoes were washed with tap water to remove foreign substances and moisture from the surface. Gelatin (GELTECH, Busan, Korea), pectin (Pectin 105, Cp Kelco Brasil S/A, Sao-Paulo, Brazil), and citric acid (RZBC Co., Ltd., Juxian, China) were used for preparing jelly. Commercial enzymes BAN480 L (alpha-amylase, 480 KNU-B/g), AMG 300 L (glucoamylase, 260 AGU/g), β-glucanase, arabanase, cellulose, hemicellulase, and xylanase complex enzyme Viscozyme L (dietary fiber degrading enzyme, 100 FBG/g) were purchased from Novozymes (Novo Nordisk Co., Copenhagen, Denmark).

### 2.2. Manufacture of Jelly

To manufacture the sweet potato hydrolysate, the washed sweet potato was divided into 2 or 3 parts (approximately 50 g) and steamed for 40 min at 100 °C. The steamed sweet potatoes were cooled to room temperature for 10 min and peeled. A paste was prepared using a blender (HY-UBF11B; Hurom, Seoul, Korea). After adding two volumes of water to the Daeyumi, Juwhangmi, and Sinjami pastes, 0.4% of alpha-amylase, glucoamylase, and dietary-fiber-degrading enzyme were each added based on the weight of the paste and reacted in a 55 °C water bath for 5 h. The supernatant was collected after centrifugation at 4000 rpm for 30 min and concentrated to 50 °Bx to prepare the saccharified concentrate. The preparation of the sweet potato jelly referred to an existing jelly manufacturing thesis, but was slightly changed [6]. The sweet potato jelly was produced by mixing 62.94% saccharified concentrate, 36.2% gelatin + water, 0.66% pectin, and 0.2% citric acid followed by heating and concentration for 10 min. The process flow is shown in Figure 1.

### 2.3. Measurement of Free Sugar Content

Free sugar content was measured using a modifying version of the method detailed by Woo et al. [11]. The samples were diluted to an appropriate concentration, filtered through a 0.45 μm membrane filter, and analyzed by HPLC (Jasco System, Tokyo, Japan). The column was a Luna 5 μm NH-2 100 Å column (4.6 mm × 250 mm ID, Phenomenex, Torrance, CA, USA). Acetonitrile: water (80:20, *v/v*) was used as the mobile phase. An ELSD detector was used (Waters 2420, Waters, Milford, MA, USA). The flow rate was 1 mL/min, and the injection volume was 20 μL. Fructose, glucose, sucrose, and maltose were used as standard products purchased from Sigma-Aldrich.

### 2.4. Evaluation of the Quality Characteristics

The moisture content was measured using an infrared moisture analyzer (XM-60, Precisa Instrument Ltd., Zurich, Switzerland) [12]. The amount of total soluble solids was measured using a refractometer (Atago hand refractometer, N1, Tokyo, Japan) calibrated with distilled water at 20 °C. pH was measured using an Orion 4 Star pH meter (Thermo Scientific, Beverly, MA, USA). The total acidity was determined as the amount of standardized 0.1 N NaOH required for neutralization, with 1% phenolphthalein being used as an indicator. The value was expressed as equivalents of acetic acid. Chromaticity was measured by repeating the L (lightness), a (redness), and b (yellowness) values three times using a colorimeter (CR-300, Konica Minolta, Osaka, Japan) standardized using a white standard plate. The standard values for the calibration of the color measurements were 96.28 for the L value, −5.06 for the a value, and 7.14 for the b value. Texture was measured using a texture analyzer (TA-XT2, Stable Micro Systems Ltd., London. UK), and hardness, adhesiveness, springiness, chewiness, gumminess, and cohesiveness were measured three times for each sample [7]. The measurement sample was cut into pieces with a diameter of 20 mm and a height of 10 mm and was measured at 20 °C. The measurement conditions for the texture analyzer were as follows: pre-test speed of 2 mm/s, test speed of 2 mm/s, post-test speed of 2 mm/s, and distance of 5 mm (50% strain). In addition, a 45 mm cylinder probe was used as an accessory.

### 2.5. Measurement of the Total Polyphenol and Flavonoid Contents

The total polyphenol content was measured as described by Dewanto et al. [13]. The sample extract (100 μL) was combined with 2 mL of 2% Na_2_CO_3_ and left at room temperature for 3 min. Thereafter, 100 μL of 50% Folin–Ciocalteu’s phenol reagent was added and reacted for 30 min in the dark. The absorbance was measured at 750 nm using a spectrophotometer (Epoch Microplate Spectrophotometer, BioTek Instruments Inc., Winnski, VT, USA). The total polyphenol content was expressed as milligrams of gallic acid equivalent (GAE) per 100 g of sample. The total flavonoid content was measured using the method described by Zhishen et al. [14]. The sample extract (250 μL) was mixed with 1 mL of distilled water and 75 μL of 5% NaNO_2_ followed by 150 μL of 10% AlCl_3_·H_2_O 5 min later. The mixture was allowed to stand for 6 min, and 500 μL of 1 M NaOH was added and left for 11 min in the dark. The absorbance was measured at 510 nm using a spectrophotometer. The results were expressed as milligram catechin equivalents (CE) per 100 g of sample.

### 2.6. Measurement of ABTS and DPPH Radical Scavenging Activity

The ABTS and DPPH radical scavenging abilities were measured using the method described by Choi et al. [15] with some modifications. ABTS (7.4 mM) and potassium persulfate (2.6 mM) were incubated in the dark for one day to form ABTS. + cations so that the absorbance value was 1.4 at 735 nm, and it was diluted with distilled water. To 1 mL of diluted ABTS. + solution, 50 μL of the extract was dissolved in each concentration, and the change in absorbance was measured after 30 min. The ABTS radical scavenging ability was expressed as equivalents of ascorbic acid. The DPPH radical scavenging activity was measured after dissolving 0.00788 g of 0.2 mM DPPH in 99.9% ethanol and adjusting the volume to 100 mL for 60 min. After 0.2 mL of the sample was added to 0.8 mL of the DPPH solution, the absorbance was measured at 520 nm using 0.2 mL of the sample at ambient temperature for 30 min. The radical scavenging ability was expressed as an IC50 value.

### 2.7. Sensory Evaluation

The sensory evaluation was performed following approval from the Institutional Review Board (IRB No. CBNU-202210-HR-0218) of the Bioethics Committee of Chungbuk National University and was conducted with reference to the paper [15]. Among the undergraduate (graduate) students at Chungbuk National University, 30 who provided informed consent were selected as the sensory panels. After selecting 30 sensory panels and conducting preliminary training, 20 of them were selected to conduct the sensory evaluation. After cutting the samples to a certain size (1 cm × 1 cm × 1 cm) for sensory testing, numbers were written on the containers of each sample, and the order of presentation was randomized. Purified water (20 ± 1 °C) and a spit cup were provided to enable rinsing of the mouth between samples. The mouth was thoroughly rinsed 2–3 times. Color, flavor, taste, texture, and overall acceptance were evaluated on a 7-point scale (very dislike: 1 point, neutral dislike: 2 points, slightly dislike: 3 points, neutral: 4 points, moderately good: 5 points, moderately good: 6 points, very good: 7 points).

### 2.8. Statistical Analysis

For the statistical analysis, the mean and standard deviation of each measurement group were calculated using the SPSS statistical program (Statistical Package for the Social Science, Ver. 18.0; SPSS Inc., Chicago, IL, USA), and the difference between the means of the measured values was independent. A sample *t*-test (Student’s *t*-test) was performed followed by one-way ANOVA to determine whether a difference existed between the treatment conditions.

## 3. Results and Discussion

### 3.1. Free Sugar Content of the Enzyme Hydrolysate

The free sugar content of the enzyme hydrolysate prepared via the enzyme treatment of the sweet potato paste is shown in Table 1. Fructose, glucose, sucrose, and maltose were identified in the sweet potato hydrolysate. The total free sugar contents of the Daeyumi, Juwhangmi, and Sinjami cultivars were 33.30, 39.56, and 39.46%, respectively, which increased to 92.14, 74.25, and 82.13%, respectively, after the enzyme treatment. During the enzyme treatment, maltose decomposed, and the glucose content increased. The fructose content ranged from 2.41 to 4.87%, indicating no difference being induced by the enzyme treatment. The glucose content increased from 1.39 to 1.93% in the control group and 59.21 to 83.49% in the enzyme treatment group. The sucrose content ranged from 3.76 to 13.88% depending on the cultivar, and no difference was recorded following the enzyme treatment. The total starch content of the Daeyumi, Juwhangmi, and Sinjami cultivars was 64.41%, 33.99%, and 43.27%, respectively [16]. When this starch is hydrolyzed by α-amylase, it is converted to maltose and then glucose by glucoamylase. Accordingly, the glucose content was thought to have increased after the hydrolysis. The free sugar content of raw sweet potatoes was detected as fructose, glucose, and sucrose. Juwhangmi had a higher sucrose content than the Daeyumi and Sinjami cultivars [17]. According to Ra [18] and Kum [19], maltose can be produced by the enzymatic hydrolysis of pregelatinized starch during heat treatment, and this maltose is not detected upon conversion to glucose by glucoamylase treatment [20]. In this study, maltose was determined to be converted to glucose by the glucoamylase treatment.

### 3.2. Quality Characteristics of Jelly

The moisture content, total soluble solids, pH, and total acidity of the jelly made from the different sweet potato cultivars are shown in Table 2. The moisture content of the sweet potato jelly ranged from 22.30 to 22.71%, and the total soluble solid content ranged from 69.00 to 69.33 °Bx, highlighting the small difference between the varieties. The Sinjami cultivar had the lowest pH (4.28), and the highest total acidity was 16.05. In terms of chromaticity, as shown in Table 3, Sinjami had the lowest lightness value of 15.96, Juwhangmi had the highest redness value of 6.54, and Daeyumi had the highest yellowness value of 9.90 (Figure 2). Among the texture profile analysis characteristics shown in Table 4, the hardness values of Daeyumi, Juwhangmi, and Sinjami jelly were 493.98, 492.52, and 500.58 g, respectively; the chewiness values were 273.01, 279.80, and 273.49 g, respectively; and the gumminess values were 440.43, 447.71, and 444.32 g, respectively, indicating no significant differences between the cultivars. In jelly studies performed with aronia juice, the hardness and chewiness of the jelly were found to increase as the amount of juice increased [21]. However, in jelly studies performed using silkworm powder, the hardness, chewiness, and stickiness decreased as the amount of silkworm powder increased [22]. Therefore, the texture of the jelly varies depending on the type and amount of ingredients added. However, in this study, the sweet potatoes were hydrolyzed enzymatically, filtered, and concentrated to manufacture the jelly, leading to a small difference between the sweet potato varieties.

### 3.3. Total Polyphenol and Flavonoid Content

The total polyphenol and flavonoid contents of the jelly from the different sweet potato cultivars are presented in Table 5. Sinjami had a high total polyphenol content of 446.14 mg GAE/100 g, while Daeyumi had a low content of 138.36 mg GAE/100 g. The Sinjami cultivar also had a high total flavonoid content of 243.59 mg CE/100 g, while the Daeyumi cultivar had a low content of 0.26 mg CE/100 g. The total polyphenol and flavonoid contents appeared in the order of Sinjami > Juwhangmi > Daeyumi cultivars. Song [23] reported that the content of total phenolic compounds according to the flesh color of sweet potato was 13.1, 6.22, and 3.02 mg/g, respectively, for purple sweet potato > orange sweet potato > common sweet potato, which is similar to the findings of this experiment. This finding was because colored sweet potatoes contain more phenolic compounds than common sweet potatoes. In particular, in purple sweet potatoes (Sinjami), the total phenol content is higher than that in common and orange sweet potatoes owing to the large amount of anthocyanin pigments [24,25,26].

### 3.4. ABTS and DPPH Radical Scavenging Activity

The ABTS and DPPH radical scavenging activities of the jellies from different sweet potato cultivars are shown in Table 6. By measuring the antioxidant activities of the Daeyumi, Juhwangmi, and Sinjami jelly, the ABTS radical scavenging activities were 6.47, 11.89, and 29.46 mg AAE/100 g and the DPPH radical scavenging activities were 5.43, 8.78, and 35.91 mg AAE/100 g, respectively. The Sinjami cultivar had the highest activity, followed by the Juwhangmi and Daeyumi cultivars. According to Huang [27], the antioxidant activity of purple sweet potato is superior to that of orange and common sweet potatoes, and the DPPH radical scavenging ability of sweet potato methanol extract is 2.5 higher in purple sweet potato than in orange and common sweet potatoes [23]. Pigments, such as the anthocyanins, beta-carotene, and phenolic substances contained in colored sweet potatoes, may affect the radical scavenging activity.

### 3.5. Sensory Evaluation

The results of the sensory evaluation of the jelly from the different sweet potato cultivars are shown in Table 6. Based on the sensory characteristics of the jelly, color was associated with a high preference in the order of Daeyumi (6.20), Juwhangmi (5.50), and Sinjami (4.65), and no difference in flavor was found among the cultivars. In terms of taste, Daeyumi had the highest preference (6.00), while Juwhangmi had the lowest preference (4.15). Daeyumi also had the highest preference (5.40) in terms of texture. Regarding overall acceptance, the scores were 6.35, 4.45, and 5.30 for Daeyumi, Juwhangmi, and Sinjami, respectively. Based on the sensory evaluation of sweet potato chips, lower values in terms of taste, texture, and overall acceptance were obtained for the Juwhangmi cultivar relative to the Daeyumi and Sinjami cultivars [28]. Although the product differs from the jelly used in this study, the product made of Juwhangmi can be predicted to be of lower quality than the other varieties because it was derived from the same raw material. Regarding the relationship between the sensory test results and quality characteristics, there were no significant differences in physicochemical characteristics and texture profiles based on the cultivars; however, the Sinjami cultivar had the best color and antioxidant activity. Regarding overall acceptance, the Daeyumi cultivar had the highest score, but considering the functionality of the Sinjami cultivar, it also had excellent results.

## 4. Conclusions

In this study, jelly was prepared by hydrolyzing sweet potato tubers without adding sugar, and the quality characteristics of the jelly prepared from various sweet potato varieties were investigated. Three cultivars of sweet potato with different flesh colors—Juwhangmi (orange color), Sinjami (purple color), and Daeyumi (yellow flesh color)—were selected to prepare a paste, enzyme-treated to prepare a hydrolyzed product, and then concentrated to prepare the jelly. The total free sugar content increased during the enzyme hydrolysis, and no difference in fructose and sucrose contents was found following the enzyme treatment. When the enzyme mixture was used, maltose was decomposed and not detected, while the glucose content markedly increased. Owing to the quality characteristics analysis of the jelly according to the sweet potato cultivars, the moisture content was 22%, the total soluble solid content was 69 °Bx, the pH ranged from 4.28 to 4.51, and the total acidity ranged from 12.13 to 16.05. In terms of chromaticity, Sinjami had the lowest lightness, Juwhangmi had a high redness, and Daeyumi had a high yellowness. There was no difference in texture among the cultivars; however, the Sinjami cultivar had the highest total polyphenol content, flavonoid content, and antioxidant activity. Our findings indicate that jelly can be manufactured by saccharifying sweet potatoes. Sweet potato jellies with various colors and functionalities can be manufactured according to the difference in the color and functional ingredient content of sweet potatoes, and the characteristics of raw sweet potatoes have a remarkable influence on the quality characteristics of the jelly.

## Figures and Tables

**Figure 1 foods-12-01585-f001:**
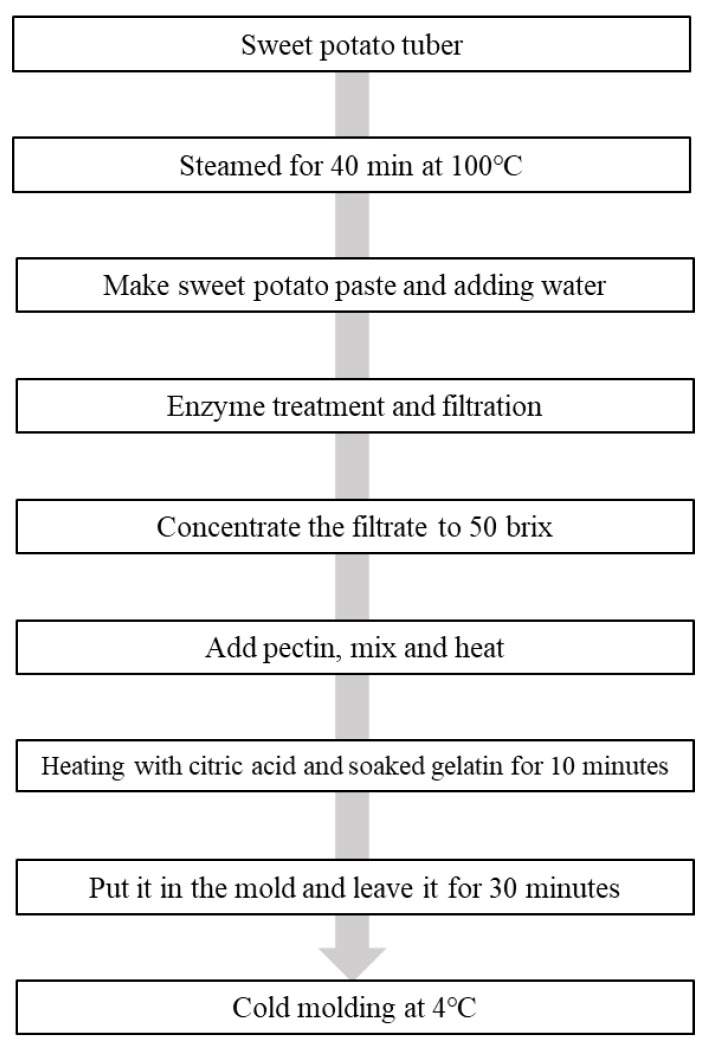
Jelly manufacturing process flow.

**Figure 2 foods-12-01585-f002:**
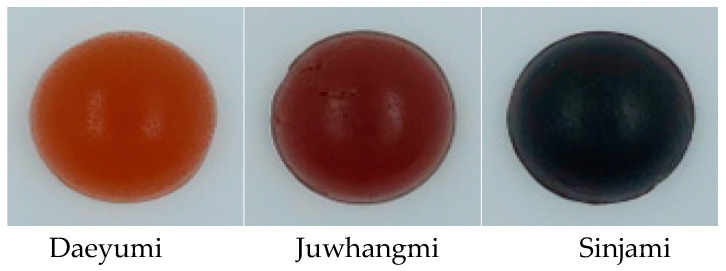
Photographs of jelly made by different sweet potato cultivars.

**Table 1 foods-12-01585-t001:** Free sugar content of enzyme hydrolysate of sweet potato pastes (dry basis %).

Cultivar	Enzyme	Fructose	Glucose	Sucrose	Maltose	Total Free Sugar
Daeyumi	No treatment	4.87 ± 0.13 ^a(1)(2)^	1.79 ± 0.10 ^d^	3.76 ± 0.18 ^c^	22.88 ± 0.10 ^b^	33.30 ± 0.15 ^e^
	Treatment ^(3)^	4.76 ± 0.02 ^b^	83.49 ± 2.67 ^a^	3.89 ± 0.24 ^c^	N.D.	92.14 ± 2.90 ^a^
Juwhangmi	No treatment	2.43 ± 0.01 ^de^	1.39 ± 0.00 ^d^	13.88 ± 1.22 ^a^	21.87 ± 0.15 ^c^	39.56 ± 1.30 ^d^
	Treatment	2.41 ± 0.01 ^e^	59.21 ± 0.30 ^c^	12.64 ± 1.26 ^a^	N.D.	74.25 ± 0.96 ^c^
Sinjami	No treatment	2.53 ± 0.08 ^d^	1.93 ± 0.14 ^d^	8.97 ± 1.00 ^b^	26.03 ± 0.13 ^a^	39.46 ± 1.29 ^d^
	Treatment	2.69 ± 0.03 ^c^	70.74 ± 0.15 ^b^	8.71 ± 0.16 ^b^	N.D.	82.13 ± 0.28 ^b^

^(1)^ Each value is expressed as the mean ± standard deviation (n = 3). ^(2)^ Different small letters in the same column indicate a significant difference by Duncan’s range test (*p* < 0.05). ^(3)^ Enzyme treatment: α-amylase, gluco-amylase, and viscozyme L complex mixture. N.D.: Not detected.

**Table 2 foods-12-01585-t002:** Moisture content, total soluble solid, pH, and total acidity of jelly with different cultivars.

Cultivar	Moisture (%)	Total Soluble Solid (°Bx)	pH	Total Acidity (%)
Daeyumi	22.65 ± 0.78 ^a(1)(2)^	69.33 ± 0.33 ^a^	4.51 ± 0.03 ^a^	12.13 ± 0.23 ^c^
Juwhangmi	22.71 ± 0.87 ^a^	69.00 ± 1.00 ^a^	4.47 ± 0.11 ^a^	14.00 ± 0.00 ^b^
Sinjami	22.30 ± 0.52 ^a^	69.00 ± 0.00 ^a^	4.28 ± 0.09 ^b^	16.05 ± 0.48 ^a^

^(1)^ Each value is expressed as the mean ± standard deviation (n = 3). ^(2)^ Different small letters in the same column indicate a significant difference by Duncan’s range test (*p* < 0.05).

**Table 3 foods-12-01585-t003:** Chromaticity of jelly with different cultivars.

Cultivar	Hunter’s Color Value		
	L	a	b
Daeyumi	22.74 ± 0.48 ^a(1)(2)^	4.21 ± 0.02 ^b^	9.90 ± 0.30 ^a^
Juwhangmi	21.31 ± 0.00 ^b^	6.54 ± 0.01 ^a^	8.85 ± 0.01 ^b^
Sinjami	15.96 ± 0.03 ^c^	−0.25 ± 0.10 ^c^	3.22 ± 0.05 ^c^

^(1)^ Each value expressed as the mean ± standard deviation (n = 3). ^(2)^ Different small letters in the same column indicate a significant difference by Duncan’s range test (*p* < 0.05). L: lightmess, a: redness, b: yellowness.

**Table 4 foods-12-01585-t004:** Texture profile analysis of jelly with different cultivar.

Cultivar	Properties					
	Hardness (g)	Adhesiveness (g.s)	Springiness	Chewiness	Gumminess	Cohesiveness
Daeyumi	493.98 ± 45.98 ^(1)^	−41.00 ± 1.95	0.61 ± 0.10	273.01 ± 76.37	440.43 ± 56.38	0.89 ± 0.03
Juwhangmi	492.52 ± 11.60	−39.83 ± 2.42	0.60 ± 0.04	279.80 ± 52.17	447.71 ± 113.49	0.89 ± 0.05
Sinjami	500.58 ± 29.68	−39.43 ± 1.65	0.60 ± 0.08	273.49 ± 27.92	444.32 ± 33.94	0.90 ± 0.03

^(1)^ Each value is expressed as the mean ± standard deviation (n = 3).

**Table 5 foods-12-01585-t005:** Total polyphenol and flavonoid content of jelly with different cultivars.

Cultivar	Total Polyphenol (mg GAE ^(1)^/100 g)	Total Flavonoid (mg CE ^(2)^/100 g)	ABTS Radical Scavenging Activity (mg AAE ^(3)^/100 g)	DPPH Radical Scavenging Activity (mg AAE/100 g)
Daeyumi	138.36 ± 2.97 ^c(4)(5)^	0.26 ± 0.00 ^c^	6.47 ± 0.29 ^c^	5.43 ± 0.31 ^c^
Juwhangmi	287.35 ± 1.98 ^b^	19.39 ± 0.00 ^b^	11.89 ± 0.02 ^b^	8.78 ± 0.29 ^b^
Sinjami	446.14 ± 0.99 ^a^	243.59 ± 9.02 ^a^	29.46 ± 0.12 ^a^	35.91 ± 1.44 ^a^

^(1)^ mg gallic acid equivalent (GAE) per 100 g. ^(2)^ mg catechin equivalent (CE) per 100 g. ^(3)^ mg ascorbic acid equivalent (AAE) per 100 g. ^(4)^ Each value is expressed as the mean ± standard deviation (n = 3). ^(5)^ Different small letters in the same column indicate a significant difference by Duncan’s range test (*p* < 0.05).

**Table 6 foods-12-01585-t006:** Sensory evaluation score of jelly with different cultivars.

Cultivar	Color	Falvor	Taste	Texture	Overall Acceptance
Daeyumi	6.20 ± 0.89 ^a(1)(2)^	5.45 ± 0.89 ^a^	6.00 ± 0.79 ^a^	5.40 ± 0.94 ^a^	6.35 ± 0.67 ^a^
Juwhangmi	5.50 ± 0.69 ^b^	5.40 ± 0.94 ^a^	4.15 ± 0.75 ^c^	4.55 ± 0.94 ^b^	4.45 ± 0.69 ^c^
Sinjami	4.65 ± 0.99 ^c^	5.40 ± 0.60 ^a^	5.20 ± 0.83 ^b^	5.00 ± 0.92 ^ab^	5.30 ± 0.73 ^b^

^(1)^ Each value is expressed as the mean ± standard deviation (n = 20). ^(2)^ Different small letters in the same column indicate a significant difference by Duncan’s range test (*p* < 0.05).

## Data Availability

The data used to support the findings of this study can be made available by the corresponding author upon request.

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
