# Peer review of "Quality Characteristics of Sweet Potato Jelly Prepared Using the Enzymatic Saccharification Method"

_foods, 2023, doi:10.3390/foods12081585_

Round 1
Reviewer 1 Report
Section 2.2 - Process flow would be helpful for the readers. Please add if the authors see relevant.
Section 2.4 - is it physical qualities?
Section 2.7 - 1cm3 - for tasting, is this a common quantity that the jelly is consumed in Korean context?
Overall acceptance on sensory panel? Can the author justify why they do this? Acceptance/hedonic measures should only be done with consumer panel?
Need more info as well on how the panel is trained?
Table 1. Isn't there an interest to see the differences in species? (4) in the same cultivar between the enzyme treatments?
Table 2+3. (2) description feels weird? Isn't it between cultivar?
Table 4. No significant differences? so clear out the letter then?
Table 6. So the authors ran a hedonic measure using n=20? This is very concerning and there should be a clear reaon why this is carried out.
Author Response
Section 2.2 - Process flow would be helpful for the readers. Please add if the authors see relevant.
The flow chart of the manufacturing process has been added to Figure 1.
Section 2.4 - is it physical qualities?
The evaluation of quality characteristics of jelly is an item that analyzes physicochemical characteristics.
Section 2.7 - 1cm3 - for tasting, is this a common quantity that the jelly is consumed in Korean context?
We randomly set the size that is easy to put in your mouth and chew.
Overall acceptance on sensory panel? Can the author justify why they do this? Acceptance/hedonic measures should only be done with consumer panel?
Need more info as well on how the panel is trained?
The initial 30 panel applicants were selected and trained by 20 panel applicants who had no sensory problems, and then the 7-point scale method was used. A reference paper was attached.
Table 1. Isn't there an interest to see the differences in species? (4) in the same cultivar between the enzyme treatments?
The duncan's range test of statistical processing has been modified from variety to column.
Table 2+3. (2) description feels weird? Isn't it between cultivar?
The duncan's range test of statistical processing has been modified from variety to column.
Table 4. No significant differences? so clear out the letter then?
There is no significant difference in the text, so I will delete it.
Table 6. So the authors ran a hedonic measure using n=20? This is very concerning and there should be a clear reaon why this is carried out.
The hedonic measurement method used a 7-point scale as a measure of the state of mind for food in a relatively simple way to assess the preference.

Reviewer 2 Report
Abstract:
The abstract should be revised,d for instance, giving the general statement "Overall, jelly may be manufactured by saccharifying sweet potatoes". It is a too-general sentence.
Introduction:
What are the previous studies state about this?
What is the present status of production of Sweet production?
Need to give clarity on the need for changes in the present production state of the jelly
Need to give the information on the saccharification needs and methods and role in this work
The Use of Sweet potato in jelly preparation carried for the first time? or some other studies are reported?
What are other raw materials to make jelly?
How Sweet potato will be the advantage here?
The flow of ideas should check, hence, work on the flow of the introduction
Materials and Methods:
The methods should be clear
The procedure of Saccharification should be present in clear
Provide appropriate citations for all the methods and procedures followed
Results and discussion
In section 2.2 is it an extruder or a blender? (lines number 74 and 75)
Line 73, "cut into 2 or 3 parts" what is the size of the chunk?
Lines 79 to 81, must clearly explain the jelly preparation with the reference,
Usage of gelatin is unusual in jelly preparation better cite the source of the procedure
Line number 89, what is this? "Fructose, glucose, sucrose, and maltose were obtained from Sigma Aldrich" they are standards?
Section 2.4 cite proper references for all
Why TA is expressed as acetic acid%?
Line 126, used "+ cations" revise
Line 128, used "+ solution" what is this?
Section 2.7. line numbers 138 to 140 revise
The methods described in section 2.7 looks non-standard. Hence rewrite
Lines 145 and 146 Give the scale range and value.
Results and discussion
Section 3.1, the discussion given on the conversion of the sugars need to discuss in deep and wider conditions.
In line 61, mentioned that ". The Daeyumi cultivar was used as the control" but the control is a usual practice. In table 1, the control is changed. It is too redundant.
The first line in table 1, is given as "unit" what is this?
Footnote of table 1, presentation is not correct. Mention in appropriate form. Check the same for all the tables.
Table 2. Sugars are different in the enzyme-treated and much higher, and reported lower what is the reason (TSS)?
Section 3.5, line 250, should be table 6 instead table 5.
line 226, table 6 title is very poor
Conclusions are too vague, better to be precise.
line 272, "sweet potato starch was hydrolyzed" no it is a sweet potato tuber that was used to hydrolyze, check once.
Some of the tables can be converted to images.
Editorial issues should consider
Language issues should focus
Typographical issues should be resolved
The manuscript looks like it is not a final draft and the authors communicated an older draft mistakenly.
I suggest the revision of a manuscript with an expert.
Author Response
Reviewer 2
Abstract:
The abstract should be revised,d for instance, giving the general statement "Overall, jelly may be manufactured by saccharifying sweet potatoes". It is a too-general sentence.
The sentence you said has been modified to the sentence below.
->“This result shows that jelly may be manufactured by saccharifying sweet potatoes, and it is confirmed that the characteristics of raw sweet potatoes have a great influence on the quality characteristics of the jelly”.
Introduction:
What are the previous studies state about this?
In the introduction, it was explained that research on various functional jellies has been conducted recently. In addition, sweet potato has high yield and excellent nutritional characteristics and functionality, so it has high utilization value, so it is thought that it will be suitable for manufacturing functional jelly.
What is the present status of production of Sweet production?
In the introduction, it was explained that sweet potatoes are the 7th most produced carbohydrate supplying food in the world.
Need to give clarity on the need for changes in the present production state of the jelly
I explained the consumption and research trend of jelly in the introduction line 45-50 by citing references.
Need to give the information on the saccharification needs and methods and role in this work
A lot of sugar is required to make jelly, and a saccharification process is essential to make jelly using sweet potatoes. The saccharification method was described in the material method.
The Use of Sweet potato in jelly preparation carried for the first time? or some other studies are reported?
Research on making jelly using sweet potatoes has been reported, but no papers on making jelly using only sweet potatoes without using sugar have been found.
What are other raw materials to make jelly?
Saccharified concentrate is the main raw material, and gelatin, pectin, citric acid, and water are added.
How Sweet potato will be the advantage here?
Because sweet potato has a high starch content, it is possible to make jelly without using sugar through the saccharification process. In addition, colored sweet potatoes have various functions and colors, so it is possible to manufacture visually and nutritionally excellent jelly.
The flow of ideas should check, hence, work on the flow of the introduction
After checking the contents of what you said, We corrected to line 19.
Materials and Methods:
The methods should be clear
The procedure of Saccharification should be present in clear
Provide appropriate citations for all the methods and procedures followed
The overall process flow chart of the jelly manufacturing method was added to line 86 and explained. The methods and enzymes used for each process were clearly explained.
In section 2.2 is it an extruder or a blender? (lines number 74 and 75)
It’s blender. We has been corrected.
Line 73, "cut into 2 or 3 parts" what is the size of the chunk?
The size of the chunk is about 50g, and related information has been added.
Lines 79 to 81, must clearly explain the jelly preparation with the reference,
A cited paper was added and the manufacturing process was explained in figure 1. (line 86)
Usage of gelatin is unusual in jelly preparation better cite the source of the procedure
Gelatin was used to make jelly with the properties we wanted.
Line number 89, what is this? "Fructose, glucose, sucrose, and maltose were obtained from Sigma Aldrich" they are standards?
Fructose, glucose, sucrose and maltose are standard products. I purchased it from Sigma-Aldrich company and used it. (line 96)
Section 2.4 cite proper references for all
Reference papers have been added. (line 101, 113 )
Why TA is expressed as acetic acid%?
We were checked and corrected the following sentence.
“The value is expressed as equivalents of acetic acid.” (line 105)
Line 126, used "+ cations" revise
Line 128, used "+ solution" what is this?
Means an ABTS solution in which cations are formed.
Section 2.7. line numbers 138 to 140 revise
The methods described in section 2.7 looks non-standard. Hence rewrite
The initial 30 panel applicants were selected and trained by 20 panel applicants who had no sensory problems, and then the 7-point scale method was used. A reference paper was attached.
Lines 145 and 146 Give the scale range and value.
I marked the scale range and value of the 7-point scale.
Results and discussion
Section 3.1, the discussion given on the conversion of the sugars need to discuss in deep and wider conditions.
Analysis was conducted at this level because it was judged that it would have been decomposed into monosaccharides or disaccharides by processing various enzymes such as alpha-amylase, glucoamylase, and dietary fiber degrading enzyme.
In line 61, mentioned that ". The Daeyumi cultivar was used as the control" but the control is a usual practice. In table 1, the control is changed. It is too redundant.
The control in table 1 was changed to no treatment because it was tested as a control of enzyme treatment.
The first line in table 1, is given as "unit" what is this?
All results were presented on a dry basis and were expressed in terms of units, but were removed because they could cause confusion.
Footnote of table 1, presentation is not correct. Mention in appropriate form. Check the same for all the tables.
All tables were checked and corrected.
Table 2. Sugars are different in the enzyme-treated and much higher, and reported lower what is the reason (TSS)?
The soluble solids content in table 2 is measured after the jelly has been prepared.
Section 3.5, line 250, should be table 6 instead table 5.
We were checked and corrected.
line 226, table 6 title is very poor
We were checked and corrected.
Conclusions are too vague, better to be precise.
We were checked and corrected. (line 293-297)
line 272, "sweet potato starch was hydrolyzed" no it is a sweet potato tuber that was used to hydrolyze, check once.
Based on what you said, I have amended the following sentence. (lines 284-286)
Some of the tables can be converted to images.
If the final version of the thesis needs revision, it will be corrected.
Editorial issues should consider
Language issues should focus
Typographical issues should be resolved
The manuscript looks like it is not a final draft and the authors communicated an older draft mistakenly.
I suggest the revision of a manuscript with an expert.
We made English proofreading in 'editage' and attached proof of proofreading.

Round 2
Reviewer 1 Report
The clarification of randomly set the size is not justified, the evaluation size should be done mimicking real life scenario, cutting it too small (or random as the author claimed) would result in changes in textural/mouthfeel percept.
There is no way that hedonic test is only being carried out by 20 people, a consumer test should be attempted with at least ~100 people or more
Author Response
Review 1
The clarification of randomly set the size is not justified, the evaluation size should be done mimicking real life scenario, cutting it too small (or random as the author claimed) would result in changes in textural/mouthfeel percept.
The reason why we set the size of 1cm3 is because it is the size of the jelly we want to release as a product. So, We answered '' We randomly set the size that is easy to put in your mouth and chew.'' last time.
There is no way that hedonic test is only being carried out by 20 people, a consumer test should be attempted with at least ~100 people or more
I misunderstood the first revision question. Our test was a analytical test, not a consumer test. Among them, the attribute difference test(7-point Likert scale) was used.

Reviewer 2 Report
All the comments are handled perfectly.
Author Response
Revision 2
All the comments are handled perfectly.
Thank you for reviewing for a good article.
